# Development and Validity Evaluation of the Index of Social Work Process in Promoting Social Participation of Welfare Recipients (SWP-PSP) in Japan

**DOI:** 10.3390/ijerph22091458

**Published:** 2025-09-20

**Authors:** Yukiko Takagi, Hideki Hashimoto

**Affiliations:** Department of Health and Social Behavior, School of Public Health, The University of Tokyo, 7-3-1 Hongo, Bunkyo City, Tokyo 113-8654, Japan; hidehashim@m.u-tokyo.ac.jp

**Keywords:** measurement development, social work evaluation, unemployment, social connection, promoting social participation

## Abstract

Social workers are required to have the capacity to effectively support welfare recipients to restore their labor participation for social inclusion. However, a systematic method for process evaluation of this capacity has not yet been established. In this study, we developed the Index of Social Work Process in Promoting Social Participation of Welfare Recipients (SWP-PSP) to address this gap. Item domains and pools were prepared by referring to existing social work guidelines and human capital management theories, and content and face validity were confirmed by an expert panel review. The initial 75 items were revised to 44. We conducted a cross-sectional survey with 139 social workers working in public livelihood support at various municipal authorities in Japan. Item response theory analysis was performed for item selection, followed by the criterion-related validity test for convergent validation using Utrecht Work Engagement (UWE) scale scores as a reference. The selected 20 items with four domains were moderately correlated with UWE scores (Pearson’s correlation coefficient r = 0.35). Certified social workers demonstrated a stronger correlation with UWE (r = 0.44) than social workers without certification (r = 0.26). Cronbach’s alpha coefficients in each domain were over 0.77. These results indicate the reliability and validity of the SWP-PSP. This measure may be helpful for the evaluation of social workers’ capacity to promote social inclusion of welfare recipients.

## 1. Introduction

Accumulating evidence suggests that a lack of social connection is associated with poor health [1]. Among people of working age, labor participation is not only necessary for financial stability but also for maintaining social contact [2,3]. Several empirical studies have reported that unemployment is associated with a decline in indicators of social participation [4,5,6].

Welfare programs in many countries provide unemployed individuals with financial assistance while requiring them to return to work to enhance their economic self-reliance. Governments in several countries have implemented welfare reforms with employment obligations for program recipients, though some studies have demonstrated that such reforms may harm recipients’ health [7]. Instead, more recent arguments call for the well-being of recipients to be the target of welfare programs including labor participation support [7,8].

Although previous studies of welfare programs or job training have investigated the effectiveness of employment support on labor, health, or behavioral outcomes [9,10,11,12,13,14], the processes underlying these programs have not been well described. Outcomes are often used as the primary index for service effectiveness, though outcomes are influenced by conditions other than program effectiveness. In contrast, process measures have been used to assess the critical steps affecting the quality of services [15]. While case management, counseling and encouragement have been suggested as effective employment support processes in the existing literature [9,13,14], to our knowledge, no systematic process evaluation frameworks exist for employment-support social work for welfare recipients. This lack of a systematic framework precludes evidence-based improvements in service quality.

Accordingly, this study has two objectives. First, to specify a standard employment support process with consideration of recipients’ well-being. Second, to develop and validate an assessment tool for this process, the Index of Social Work Process in Promoting Social Participation of Welfare Recipients (SWP-PSP).

## 2. Materials and Methods

### 2.1. Domain and Item Development

We conducted a review of existing literature investigating the employment support process and social work competencies. We referred to the National Occupational Standards for Social Work [16] in the UK, the Subject Benchmark Statement [17] in the UK, Educational Policy and Accreditation Standards [18] in the United States, and Practical skills required for certified social workers [19] in Japan because these standards are widely cited in standard setting of social work. We extracted six concepts commonly included in these quality standards, namely: (1) interpersonal relationship building; (2) support for appropriate decision-making; (3) flexibility for changes; (4) development and coordination of social resources; (5) provision of tailored services or information; (6) helping build the confidence of service users by specifying their strengths. Given that both (3) and (5) address adaptable skills in response to contexts, we combined them, resulting in five concepts extracted from the social work field.

We also referred to theories in human resource management for enhancing employees’ motivation and performance, which we considered would provide a technical rationale for the process of social workers helping recipients. By referring to Transformational Leadership [20], Leader–Member Exchange theory [21,22,23,24], Self-Determination theory [25,26], and Goal Setting theory [27,28], we extracted five concepts related to social work process, namely: (a) trust building, (b) active listening, (c) suggesting goal options, (d) helping gain confidence, and (e) timely information provision. Since (a) trust building inevitably requires (b) active listening, we recognized them as conceptually inseparable, resulting in four concepts.

Finally, we integrated these extracted concepts into five domains, namely: Effective Relationship, Deliberative Support, Positive Feedback, Tailored Information, and Network Development. The conceptual integration is described in Appendix A.

We prepared item pools using the extracted domains and referring to 43 existing measurement batteries in studies of leadership, coaching, psychological counseling, career counseling, and business networking. We searched PubMed and Web of Science with keywords related to the five domains, and prepared an item pool of 75 items for further discussion. The five domains and examples of item drafts are shown in Appendix A.

### 2.2. Content and Face Validity

We recruited experts during March–May 2024 using snowball sampling. The inclusion criterion was being an academic researcher or practitioner of Public Livelihood Support (PLS) social work with years of practice experience as a PLS social worker. Five eligible experts (four academic researchers and one ex-PLS social worker) consented to participation in this study. All of them had years of experience in local welfare offices. Three were serving in supervisory positions. Two of them had experience as policy advisors for central and local governments on welfare reform and civil worker training. We obtained experts’ feedback through an online focus group discussion on 22 August 2024, followed by feedback via e-mail.

### 2.3. Reliability and Criterion-Related Validity

The cross-sectional survey was conducted from November 2024 to January 2025. Snowball sampling was implemented through the four experts’ professional networks to recruit PLS social workers. In addition, we recruited respondents from two municipal welfare offices in the greater metropolitan area for the survey. The inclusion criterion was being a current or ex-PLS social worker. Participants anonymously completed the online questionnaire after providing informed consent at the beginning of the survey. The survey asked participants to reflect their usual support process with case recipients. All items of the SWP-PSP were scored using a 6-point rating ranging from 1 (“never”) to 6 (“always”). Additionally, we obtained information about participants’ gender, age, years of experience as a PLS social worker, and certification status. In the Japanese PLS system, the required services and processes related to welfare support are identical for practitioners regardless of their social work certification status only if they receive a short-term training, while the national licensure for social workers requires specialty educational credits for licensure eligibility. Given this context, we used licensure status as a proxy marker of PLS social workers’ specialty educational experience when discerning whether any differences existed between self-rated performance scores.

### 2.4. Data Analysis

We applied item response theory (IRT) analysis to estimate the discrimination and difficulty for each item. Because the distributions of each item were skewed, we employed conventional dichotomous IRT, in which option 6 (always) was correct (=1) and all other options were incorrect (=0). After the binary transformation, we tested unidimensionality using confirmatory factor analysis (CFA) and exploratory factor analysis (EFA), local independence based on Yen’s Q3 statistic [29] checking the residual correlation matrix of the CFA, and monotonicity using the Mokken Scale Procedure (MSP) [30]. After excluding a domain that did not meet these assumptions, IRT analysis (two parameters) was conducted in each domain. Discrimination parameters higher than 1.7 are taken as very good according to Baker and Kim [31], but clear criteria for difficulty do not exist. However, items with difficulty below 0 are considered too easy [32,33]. Therefore, we decided to select items with higher discrimination (>2.0) and an acceptable range of difficulty (>0.1). We then conducted criterion-related validation using the total score of adopted items. We used the Utrecht Work Engagement (UWE) scale [34] as a reference because UWE reflects an individual’s recognition of the meaningfulness of their work [35,36] and relates to work performance [37], which we expected would correlate with the self-evaluation of the process quality of employment support for welfare recipients. Higher SWP-PSP would be expected to predict better social work, which would be expected to be correlated with higher UWE scores. We investigated Pearson’s correlation coefficient with UWE for all participants, as well as for participants stratified by certification status, to examine the differences. For reliability testing, Cronbach’s alpha coefficients were calculated in each domain. All analyses were performed using Stata version 17 software (StataCorp, College Station, TX, USA).

## 3. Results

### 3.1. Content and Face Validity

The expert panel reached consensus on the significance of relationship building through supportive and open communication, with tailored information to help enhance self-reliance. They also agreed on the importance of resource networking and motivating skills, and that technocratic enforcement of labor participation may not work. The panel advised the revision of domain names and item wording for clarity, as detailed in “The process of item revision in the focus group interview” in the Appendix A. Subsequently, the pilot measure with 44 items in five domains (Development of Effective Relationship, Support for Deliberation, Positive Feedback, Provision of Tailored Information, and Network Development) was prepared for further validation testing.

### 3.2. Reliability and Criterion-Related Validity

#### 3.2.1. Demographic Characteristics of Cross-Sectional Research

The questionnaire was completed by 139 PLS social workers. Participants’ average age was 35.7, and their years of experience were approximately divided equally into three categories. The certified social workers comprised 43.2% of participants. The mean score of certified social workers was significantly greater than that of social workers without certification by 10.75 points (Wilcoxon rank-sum test: z = −2.71, *p* = 0.007). The mean scores for each domain were 5.31, 4.77, 4.86, 5.02, and 4.83, respectively (Table 1).

#### 3.2.2. Item Selection by Item Response Theory Analysis

The CFA of the five domains resulted in a root mean square error of approximation of 0.078, standardized root mean square residual of 0.08, and comparative fit index value of 0.718. The root mean square error of approximation and standardized root mean square residual values were acceptable, but the comparative fit index value was not sufficient [38]. We then implemented EFA to examine whether the eigenvalues and the contributions of the first factor were sufficiently large in each domain. The first and second factors were similar in domain 1. Thus, the unidimensionality of domain 1 was not confirmed. The maximum value in residual correlation of CFA between five domains was 0.044; this value was close to zero, confirming local independence [29]. MSP showed that domain 1 did not converge. Therefore, we determined that monotonicity in domain 1 was rejected. Although one item of domain 2 remained, other items were selected in one scale (H = 0.48). In domains 3 to 5, all items were selected and the H coefficient was greater than 0.5 [30]. Monotonicity was confirmed in domains 2 to 5. Given that the unidimensionality and monotonicity in domain 1 were not confirmed, we decided to omit IRT analysis of that domain. IRT analysis was conducted in each of the remaining four domains, and we checked the discrimination and difficulty of each item. The number of items with discrimination >2.0 and difficulty >0.1 was 6 in domain 2 (the item not selected in the MSP was excluded), 5 in domain 3, 3 in domain 4, and 6 in domain 5. The total number of included items was 20 (Table 2).

#### 3.2.3. Criterion-Related Validity

The correlation coefficient between the non-binary total score of selected SWP-PSP items and UWE score was calculated. The coefficient was 0.35, demonstrating a moderate correlation [39] (Figure 1). When the total scores were stratified into certified social workers and social workers without certification, the coefficient was 0.44 for certified workers, and 0.26 for those without certification. We examined the measurement invariance across the two groups. The results indicated that the factor loadings were similar between those participants with and without certification, suggesting metric invariance. However, scalar invariance could not be confirmed because of the limited sample size. Although the correlations with UWE scores by domain were investigated, no explicit changes from total scores were found. Additionally, there was no correlation between years of experience and UWE scores (r = 0.10).

#### 3.2.4. Reliability

The Cronbach’s alpha coefficients for domains 2–5 were 0.80, 0.86, 0.77, and 0.84, respectively.

## 4. Discussion

Although CFA showed a weak model fit between five domains, EFA confirmed unidimensionality in each domain except for domain 1. Twenty items with discrimination >2.0 and difficulty >0.1 were selected from the four domains using IRT analysis. The correlation coefficient between SWP-PSP and UWE was moderate according to Cohen’s criterion [39]. Certified social workers with specialized education in social work demonstrated a stronger correlation with UWE. Cronbach’s alpha reached the acceptable range [40]. Thus, the current results support the reliability and validity of the SWP-PSP.

Domain 1 failed to exhibit unidimensionality in EFA and monotonicity in MSP—aspects that require further consideration. The small difference in scores of the domain items between respondents indicated a ceiling effect, presumably because basic communication was being measured. However, we still consider the concept of the domain per se remains important and essential for social work process. Further research is needed for better operationalization of the domain items.

PLS social workers without certification exhibited a weaker correlation with UWE scores compared with certified social workers. Because most of these workers were originally office clerks, they may evaluate their own performance for goals other than effective support for social participation of welfare recipients (e.g., efficiently completing clerical work). Current PLS social worker training is typically focused on the operation of PLS law and systems, with less emphasis on the ultimate goal to achieve recipient’s well-being. Thus, it may be helpful to develop training programs for PLS social workers in early career and evaluate the training process using the SWP-PSP.

Several limitations of this study should be considered. First, performance assessment of the developed scale provides evidence of measurement reliability, and construct and convergent validity. However, predictive validity has not yet been demonstrated with outcomes such as the rate of work return and health improvement among program recipients. Furthermore, the scale still needs to demonstrate proof of measure responsiveness to performance change. Second, although we relied on experts with rich practice experience and institutional knowledge, the applicability of the measure remains to be shown with a wider range of PLS social workers with diverse skills and experiences. A larger probabilistic sample could also overcome the issues of potential no-response bias and inter-group comparability with scalar invariance. Participants completed the web survey anonymously, which meant that detailed information about their backgrounds was not obtained. Third, we still need to refine the operationalized items to realize the domain concept related to effective relationship building. The application of the scale in the real-world quality management of PLS social work services would benefit from addressing these remaining issues in future research. Fourth, the current form of the SWP-PSP was designed to reflect the welfare service practice in the Japanese context. Although we expect it to be usable across Japan, further efforts to refine relevant domains may be necessary for cross-country comparisons of different political and institutional settings. The detailed presentation of the measure development processes described in this study could help researchers interested in cross-country adaptations of the scale concepts. We believe the comparative scale development will facilitate research to identify quality leverages of welfare programs and related policies to be reformed for recipients’ wellbeing and social inclusion through labor participation.

Despite the study limitations, to our knowledge, the SWP-PSP is the first evaluation measure with rigorous item selection designed to systematically assess the process of employment support for welfare recipients. With further refinement and validation of the measure could support social workers’ self-reflection and growth of their skills. It could also advance systemic skills training and evaluation of early career social workers.

## 5. Conclusions

In this study, we developed the SWP-PSP as a measure of social workers’ capacity for helping welfare recipients resume their social participation and work for well-being purposes. Evidence of the scale’s measurement reliability and validity was provided. With further refinement, the measure could serve as a tool for the empirical assessment of social work processes for service quality improvement and professional growth of social workers.

## Figures and Tables

**Figure 1 ijerph-22-01458-f001:**
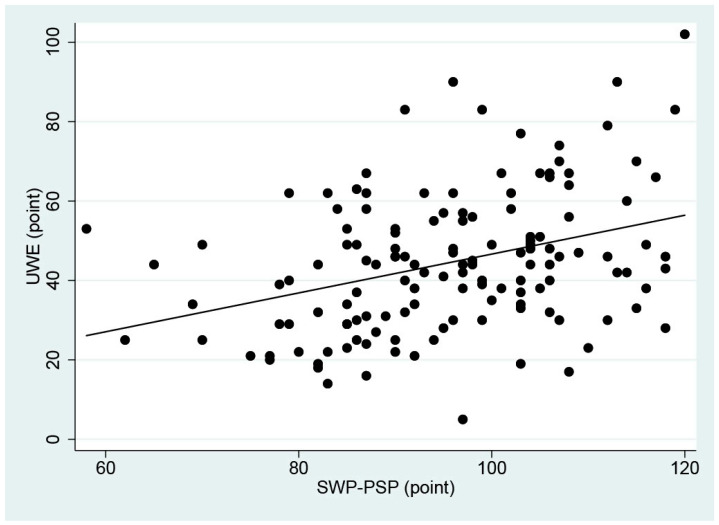
Correlation between the Index of Social Work Process in Promoting Social Participation of Welfare Recipients (SWP-PSP) scores and Utrecht Work Engagement (UWE) scores. Maximum scores were 120 (SWP-PS) and 102 (UWE). The correlation coefficient was 0.35.

**Table 1 ijerph-22-01458-t001:** Demographic characteristics of participants.

		With Certification	Without Certification
Gender			
	Male	29	60
	Female	30	18
		(Gender not reported for two participants)	
Age			
	20 s	17	27
	30 s	19	26
	40 and over	24	26
Years of experience			
	Less than 3 years	14	27
	3–5 years	20	27
	5 years or more	26	25
Mean SWP-PSP score		217.61	
(Maximum score: 264)		223.72	212.97

**Table 2 ijerph-22-01458-t002:** Final items and item response theory (IRT) parameters.

Items	Discrimination	Difficulty
Domain 2		
I explain the meaningfulness of work to the client by describing benefits of social participation.	2.30	1.02
I ask the client about their worries or anxieties related to work.	2.27	0.52
I ask the client about their own future vision.	2.10	0.87
I summarize and share the client’s anxieties related to work in the conversation.	2.50	0.80
I explicitly share with the client their anxieties related to work.	3.49	0.45
I suggest alternative options of work for diverse social participation depending on the client’s needs.	2.92	1.16
Domain 3		
I verbalize the client’s strengths and feedback clearly.	3.98	1.10
I encourage the client to utilize their strengths.	2.14	1.20
I encourage the client by telling them that I am on their side.	5.59	0.33
I acknowledge and commend the client for their efforts.	4.01	0.31
I commend the client when they make progress even if it is small.	3.09	0.34
Domain 4		
I read the client’s expressions regarding their understanding of the documents or explanations.	5.56	0.31
I read the client’s expressions regarding whether they have unspoken worries.	2.72	0.82
I confirm with the client whether they understand what they are expected to do in the procedure.	2.74	0.57
Domain 5		
I can suggest medical or social services the client can use.	2.05	1.38
I know which medical or social services the client has used.	2.58	0.68
I regularly share the client’s information with their service providers.	2.27	1.44
I share problems and discuss them with the clients’ service providers.	2.67	0.78
I cooperate with the service providers to resolve their problems related to the client within my responsibility.	2.86	0.27
I discuss ways to fill the client’s service gap with their service providers.	2.54	1.21

## Data Availability

The data presented in this study are available on request from the corresponding author due to privacy rules.

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
