# Peer review of "Development and Validity Evaluation of the Index of Social Work Process in Promoting Social Participation of Welfare Recipients (SWP-PSP) in Japan"

_ijerph, 2025, doi:10.3390/ijerph22091458_

Round 1

Reviewer 1 Report

Comments and Suggestions for Authors

Dear authors,
Below are my contributions to your work.

Title and Abstract

The title is clear and accurately reflects the content of the article. However, it may be improved by including the geographic context (Japan), to help situate the reader. The abstract presents the objective, methodology, and results clearly. It would be helpful to specify how many items were initially generated and how many were selected for the final version of the instrument. Also, consider explicitly mentioning the use of the Utrecht Work Engagement Scale (UWE) as a criterion for convergent validation.

Introduction

The introduction clearly establishes the relevance of the topic and the lack of standardized tools to evaluate the social work process aimed at supporting labor participation among welfare recipients. It provides solid references on the relationship between unemployment, social exclusion, and health. To strengthen this section, it is recommended to improve the argumentative transition from the identified problem to the proposed solution (the SWP-PSP index). Additionally, a brief justification for choosing the UWE as a validation criterion would be beneficial.

Methodology

Item Development

The item generation process is grounded in interdisciplinary literature, which is a strength. However, more detail is needed on how the identified domains were operationalized and whether any specific conceptual framework from social work guided the item construction. It would also be appropriate to include an annex or table showing representative items by domain.

Content and Face Validity

Using an expert panel was appropriate, but it is advisable to specify the selection criteria, professional profiles, and fields of expertise of the experts. Furthermore, explain how the experts’ comments were incorporated into the revised version of the instrument and whether any rating scales were used in the qualitative validation process.

Data Collection

The snowball sampling strategy should be described in more detail: how many initial nodes were used, what inclusion criteria were applied, and how many municipalities participated. It would also be helpful to justify the sample size with a power estimation for IRT and correlation analyses.

Statistical Analysis

The use of CFA, EFA, unidimensionality tests, and Item Response Theory (IRT) is appropriate. However, the decision to dichotomize the 6-point response scale should be justified, as this could imply a loss of information. Consider discussing why a graded response model was not used. Further detail is also needed on the criteria for item selection (e.g., thresholds for discrimination and difficulty) and on the rationale behind the exclusion of domain 1.

Results

The results are well organized and include key psychometric and correlation analyses. The exclusion of domain 1 is justified but could be discussed further—was item redefinition or domain splitting considered before exclusion? A summary table listing the final selected items, their domains, and IRT parameters would enhance clarity. The score differences between certified and non-certified workers should be accompanied by statistical tests (e.g., t-test or ANOVA).

Scatterplots could be improved by including labeled axes, trend lines, and brief explanations in the captions.

Discussion

The discussion properly interprets the results, including the rationale behind the weak CFA model fit, which is justified based on the index nature of the instrument. The discussion could be enriched by further exploring why non-certified workers showed weaker correlations with UWE, possibly including cultural, organizational, or institutional factors. Potential applications of the instrument in international or similar social service contexts could also be considered.

Additionally, the text could benefit from a more critical reflection on the strengths and limitations of the IRT analysis and suggestions for future research, such as longitudinal validation, sensitivity to change, or predictive validity.

Conclusions

The conclusions are consistent with the study’s objectives and findings. To increase the practical utility of the article, it is suggested to add practical recommendations on how to implement the instrument in training programs for new social workers or as a tool for continuous professional development.

Formatting, Tables, and Figures

The overall formatting is adequate. However, a summary table with selected items, their domains, and key psychometric values would enhance comprehension. Figures should be more explanatory, with visual annotations and improved clarity. A conceptual model diagram illustrating the social work process assessed by the index would also be a valuable addition.

References

The cited references are relevant, but some are outdated. It is advisable to incorporate more recent literature (last 3–5 years) on competency assessment in social work and psychometric validation in social settings. Ensure uniformity and adherence to the journal's citation style throughout the reference list.

Final Remarks

This article addresses a critical need in social work evaluation and presents a robust methodology. To strengthen its scientific value and publication potential, it is recommended to expand methodological details, clarify statistical decisions, improve results presentation, and enrich the discussion with practical applications and future validation strategies.

Comments on the Quality of English Language

Observations on English Writing Quality

1. Clarity and Conciseness

  • Several sentences are overly long or contain redundant phrases. Aim to shorten and simplify sentence structures to improve readability.

    • Example:

      "Because most PLS social workers are originally administrative workers in municipal authorities who receive only short-term training..."
      Could be revised to:
      "Most PLS social workers are former administrative staff with limited training..."

2. Word Choice and Register

  • At times, informal or vague language is used in a scientific context. Prefer precise, formal terminology.

    • "We obtained experts’ opinions..." → Consider: "Expert feedback was collected..."

    • "We then implemented..." → Prefer: "Subsequently, we conducted..."

3. Verb Tense Consistency

  • The manuscript inconsistently shifts between past and present tense. For methodological descriptions and results, use past tense consistently.

    • "We ask participants about..." → should be "We asked participants about..."

    • "All analyses are performed using..." → should be "All analyses were performed using..."

4. Article Use (a/an/the)

  • There are occasional issues with the use of definite and indefinite articles, which may obscure meaning.

    • "the discrimination and difficulty for each item" → consider "discrimination and difficulty parameters for each item"

    • "supporting paid work participation involved encouraging recipients..." → could benefit from "the support for paid work participation involved encouraging..."

5. Prepositions and Collocations

  • Some preposition choices are non-idiomatic or ambiguous.

    • "support process" → more natural: "supportive process" or "process of support"

    • "motivate the consultee workers" → could be more idiomatically expressed as "motivate the workers receiving support"

6. Repetition and Redundancy

  • Some phrases repeat the same idea unnecessarily.

    • "support recipients in (re)gaining participation in paid work..." → "in regaining paid employment" is simpler and clearer.

    • "support for deliberation by general information provision and dialogue" → consider simplifying or splitting into clearer parts.

7. Punctuation

  • There are missing or misused commas in complex sentences.

    • "We modified the item pool and asked individual experts..." → better with a comma: "We modified the item pool, and asked individual experts..."

8. Passive vs. Active Voice

  • The manuscript mostly uses active voice, which is acceptable, but in some cases, passive constructions could enhance formality and focus on the procedure.

    • "We conducted a review..." → acceptable
      But for objectivity: "A review of the literature was conducted..." is more formal.

Overall Recommendations

  • Consider a thorough copy-editing by a native English speaker or professional language editor with experience in academic writing.

  • Use tools such as Grammarly Premium or Writefull for Research for real-time language suggestions aligned with scientific norms.

  • Ensure consistency of terminology throughout the manuscript, particularly when referring to domains, item types, and statistical procedures.

Reviewer 2 Report

Comments and Suggestions for Authors

The abstract (Lines 11–28) presents a general overview of the study but lacks a strong rationale for why the SWP-PSP instrument was necessary in the first place. While the methodological steps are summarized, the justification for selecting the Utrecht Work Engagement (UWE) scale as the criterion measure is insufficient. This decision is critical to the study’s methodological soundness, yet it is neither theoretically explained nor contrasted with other potential outcome measures more directly related to social work performance or social participation outcomes. Clarifying why UWE was prioritized over more proximal or domain-specific indicators would strengthen the abstract and the study’s framing.

In the introduction (Lines 30–71), the authors repeatedly assert that no standardized methods currently exist for evaluating the effectiveness of holistic self-reliance assistance among welfare recipients. However, this claim is made without presenting a comprehensive or systematic review to substantiate it. Without such evidence, the claim risks being perceived as an overstatement. Furthermore, in Lines 49–54, the authors imply that PLS social workers with only short-term training inherently lack sufficient capacity to support holistic self-reliance. This is an assumption that may be biased unless supported by prior empirical studies. Presenting relevant quantitative or qualitative evidence would provide a more balanced and credible foundation for the argument.

The item development process (Lines 73–93) identifies five conceptual domains, yet the description of how these domains were derived lacks transparency. The literature review process is not described in sufficient detail—there is no indication of search scope, inclusion criteria, or whether multiple reviewers independently coded and agreed on domain definitions. Without a rigorous content mapping procedure or inter-rater reliability checks, the process of finalizing the domains appears ad hoc. The study would benefit from a clearer explanation of how initial conceptual frameworks were validated prior to expert review.

In the section on content and face validity (Lines 94–102), the selection of only five experts through snowball sampling poses a potential selection bias, as it may limit the diversity of perspectives in the review process. The absence of representation from broader professional associations or a more heterogeneous group of practitioners could reduce the objectivity of the face validity assessment. Justifying this sampling choice and explaining how diversity of perspectives was ensured would strengthen the credibility of this stage of development.

The methodology for reliability and criterion-related validity (Lines 103–137) raises further concerns. Snowball sampling for the main survey undermines the representativeness of the findings, and there is no analysis of potential non-response bias. More critically, the transformation of the original six-point Likert scale into a dichotomous variable (Line 121) likely results in substantial loss of information and reduced measurement sensitivity. This choice is not sufficiently justified, especially given the availability of polytomous IRT models, such as the graded response model, which would preserve ordinal information and likely yield more nuanced results.

The results from the IRT analysis (Lines 157–175) indicate that Domain 1 was excluded due to lack of unidimensionality and monotonicity, attributed to a ceiling effect. However, this decision is discussed only briefly and without exploring whether revising or rescaling the items could have preserved the domain. Given that Domain 1 represents fundamental communication skills—a critical aspect of social work practice—its removal raises conceptual concerns about the comprehensiveness of the final instrument. The authors should consider whether exclusion was the most appropriate course of action or whether item modification could have resolved the issue.

The criterion-related validity findings (Lines 176–183) report a correlation of 0.35 between the SWP-PSP and UWE scores, described as “moderate.” While statistically defensible, this magnitude may be insufficient for practical decision-making, depending on the intended use of the tool. The manuscript does not discuss what constitutes an acceptable level of validity for an instrument of this type, nor does it compare the observed correlation with benchmarks from similar measures. Including such comparisons would contextualize the significance of these results.

In the discussion (Lines 196–240), the explanation for the weak CFA model fit relies on a conceptual distinction between an “index” and a “scale” (Lines 208–212), but this justification is not well supported by empirical evidence or domain-specific literature. Furthermore, the limitations section does not address the absence of measurement invariance testing between certified and non-certified social workers, even though differences in correlation magnitudes suggest this may be relevant. Additionally, there is no exploration of the instrument’s potential cross-cultural applicability or generalizability beyond the Japanese PLS context, which is important if the authors intend the SWP-PSP to be adopted internationally.

the conclusion (Lines 241–245) is overly assertive in claiming successful validation of the SWP-PSP without adequately acknowledging the structural limitations of the study—most notably the removal of a theoretically important domain, the cross-sectional design, and the lack of broader sampling. It remains unclear whether the instrument is ready for immediate policy implementation or whether further longitudinal and cross-cultural validation is required. Explicitly outlining the next steps for research and validation would result in a more balanced and cautious conclusion.

while the manuscript addresses an important gap in the evaluation of social work processes, methodological transparency, sampling rigor, and analytic justifications require strengthening. A more systematic literature foundation, reconsideration of item scaling decisions, preservation or revision of conceptually critical domains, and additional validity testing would substantially enhance the scientific soundness and practical utility of the SWP-PSP.

Reviewer 3 Report

Comments and Suggestions for Authors

1. Title and Abstract
The title clearly reflects the purpose and scope of the study and appropriately highlights the construct being developed (SWP-PSP). The abstract is generally well-structured, summarizing the objectives, methodology, and key findings. However, there is a typographical error in “Peason’s correlation,” which should be corrected to “Pearson’s correlation.” Additionally, the abbreviation “UWE” should be introduced only once in parentheses after its full form and used consistently thereafter. Revise the abstract to correct minor errors and enhance clarity and consistency in terminology.

2. Theoretical Framework and Literature Review
The study addresses a relevant gap in evaluating social workers’ capacity to promote social participation among welfare recipients. The integration of human capital management into the social work process is an innovative perspective. However, the literature review section could be enhanced by offering a more detailed justification for the need for the SWP-PSP index and by discussing existing measurement tools and their limitations more comprehensively. Expand the theoretical grounding by deepening the discussion of prior research and clarify the added value of this new index in comparison to existing tools.

3. Methodology
The methodology is robust and well-articulated. The cross-sectional design, sample size (n=139), and the inclusion of certified vs. non-certified workers contribute to the reliability of findings. The use of Item Response Theory (IRT) for item selection and Cronbach’s alpha for internal consistency adds rigor. However, the details regarding the expert panel for content and face validity (e.g., number of experts, qualifications) are insufficiently described. Provide more information about the expert panel's composition and the procedures followed for content validation. Clarify the rationale for selecting the Utrecht Work Engagement (UWE) scale as the criterion measure and discuss its theoretical relevance to the construct.

4. Findings
The results indicate a reliable and valid four-domain, 20-item scale. Internal consistency scores (Cronbach’s α > 0.77) are acceptable. The moderate correlation with UWE (r = 0.35) supports criterion-related validity, and the higher correlation among certified workers (r = 0.44) is an insightful finding. However, the practical interpretation of these correlation coefficients is not fully explored. Discuss the implications of moderate correlations in terms of practical significance and how they inform the utility of the SWP-PSP index in real-world social work settings.

5. Discussion and Conclusion
The discussion section aligns with the findings and suggests the SWP-PSP index is promising for standardizing evaluations in social work. However, further elaboration is needed regarding the generalizability of the results. It remains unclear whether cultural or contextual factors may limit broader application. Include a more critical discussion of the limitations (e.g., sample restricted to Japan, potential response bias), and suggest how the index might be adapted or validated in different contexts or countries.

6. Overall Evaluation
Addresses a relevant and underexplored area in social work practice. Methodologically sound with multi-level validation strategies. Offers a practical tool for assessing professional capacity in promoting social participation. Literature review lacks depth in justifying the theoretical basis of the new scale. Limited detail on expert panel procedures. Interpretation of correlation findings could be expanded.

Reviewer 4 Report

Comments and Suggestions for Authors

This paper, “Development and Validity Evaluation of the Index of Social Work Process in Promoting Social Participation of Welfare Recipients (SWP-PSP), presents a methodologically rigorous and impressively detailed account of a measurement development study. The authors are to be commended for their systematic and transparent approach. The core strength lies in the sophisticated psychometric validation process. The manuscript clearly outlines a multi-phase approach, beginning with item development grounded in a literature review, followed by content and face validity checks with a panel of relevant experts. The subsequent quantitative validation is robust, employing advanced statistical techniques including Item Response Theory (IRT) analysis for item selection and both Confirmatory and Exploratory Factor Analysis (CFA/EFA) to test the instrument's underlying structure. Furthermore, the authors appropriately test for criterion-related validity by correlating the new measure (the SWP-PSP) with an established, relevant scale (the Utrecht Work Engagement scale). The discussion section is also a high point, as the authors astutely interpret complex findings, such as providing a sound rationale for the exclusion of "Domain 1" due to a likely ceiling effect and addressing the weak CFA model fit by correctly distinguishing between a psychometric "index" and a "scale". The writing is clear, well-organized, and follows a logical progression, making the complex analyses accessible to the reader.

While there is much to like about this paper, there are several concerns I have with this manuscript:

The most significant and critical concern is the manuscript's fundamental lack of fit with the International Journal of Environmental Research and Public Health. While the study is methodologically sound, its topic is highly specialized and does not align with the journal's aims and scope. The paper's focus is on the development of a professional practice tool for social workers in Japan's public livelihood support system. While social inclusion is a social determinant of health, the core contribution here is to the fields of social work evaluation, psychometrics, and social policy, not public health or environmental science. There is no component of "Environmental Research" in the study, and its connection to "Public Health" is tenuous at best. A reader of this journal would not expect to find an instrument validation study so narrowly focused on the specific processes of one profession. The implications and conclusions are directed at social work training and practice, not at public health practitioners or environmental researchers.

Furthermore, the framing of the problem in the introduction contains conceptual weaknesses. The authors correctly state that "most PLS social workers are municipal office clerks rather than specialists" who "receive only short-term training". They also define the holistic concept of self-reliance in PLS as comprising three aspects: managing one's health, participating in society, and engaging in paid work. However, the paper would be strengthened by a more thorough discussion of how the lack of specialized social work training specifically impedes the workers' ability to support these three interconnected aspects of self-reliance. The paper then makes a significant logical leap by stating, "As a result, the employment rate of welfare recipients under 65 years old who live alone without serious disabilities or diseases is reported to be only approximately 30%". This assertion implies a direct causal link between insufficient worker training and a low employment rate without providing any evidence to substantiate it. This connection is weak and overlooks a host of other systemic, economic, or client-based factors that influence employment outcomes.

A secondary concern relates to the sampling methodology and the subsequent claims of the instrument's utility. The authors utilized a snowball sampling technique to recruit participants, which they rightly acknowledge as a limitation. However, the implications of this non-probability sampling method may be understated. The authors conclude that the SWP-PSP "may be helpful for standardizing social workers' capacity regardless of certification or experience" . The goal of standardization requires that the validation sample be highly representative of the target population of social workers. A convenience sample, which may overrepresent social workers who are more engaged or accessible through the researchers' networks, limits the generalizability of the findings and, therefore, weakens the claim that this tool is ready to serve as a standardizing instrument for the profession at large.

Building on this point, a core element of the study's identity is questionable. The instrument is named the "Index of Social Work Process...", yet the sample used for its validation largely consists of individuals whom the authors distinguish as not being certified social workers. According to the data presented, 78 of the 139 participants were "social workers without certification". This means that nearly more than half of the sample used to validate a "social work" process scale are not, by the authors' own distinction, formally trained social workers. This is a complete misnomer and creates a fundamental disconnect that challenges the instrument's construct validity. Is the scale truly measuring a "social work process," or is it measuring the welfare support process of municipal clerks? The acknowledgment of this foundational issue seems to largely disqualify the validity and utility of the study's findings when framed as a social work tool.

A third concern lies with the conceptual basis for the criterion-related validation. The authors chose the Utrecht Work Engagement (UWE) scale, which measures the social worker's own enthusiasm for their job, as the proxy for effective practice. The rationale – that social workers who follow the SWP-PSP process will be more engaged – is an indirect and potentially tenuous link. A stronger validation would involve correlating the SWP-PSP scores with more direct measures of performance, such as client outcomes (e.g., rates of employment, measures of well-being, or social participation) or observational ratings of the social workers' practice. Correlating one self-report of a worker's process (SWP-PSP) with another self-report of their internal state of engagement (UWE) introduces a potential conceptual circularity and does not fully validate the instrument's effectiveness in achieving its stated purpose: "promoting social participation of welfare recipients".

Methodologically, there are two additional points of concern. First, the decision to dichotomize the 6-point Likert scale responses for the Item Response Theory (IRT) analysis is questionable. By coding only the top response ("always") as correct and all five other distinct responses as incorrect, a significant amount of statistical information and response variance is lost. More appropriate polytomous IRT models exist precisely for this type of ordinal data and would have provided a more nuanced analysis of item characteristics without discarding valuable data. Second, the study relies exclusively on self-report data for both the new instrument (SWP-PSP) and the criterion variable (UWE). This raises a significant risk of common method bias and social desirability bias, as practitioners are likely to report their professional actions and attitudes in a favorable light. Without objective data, such as observational assessments of practice or, most importantly, client-reported outcomes, the validity of the findings is inherently limited by this single-source, self-report design.

In sum, the authors have produced a high-quality, methodologically sophisticated paper that represents a valuable contribution to the field of social work. The development and validation of the SWP-PSP are executed with a commendable level of rigor. However, the manuscript is not a suitable candidate for publication in the International Journal of Environmental Research and Public Health due to a significant misalignment with the journal's scope. The work is far better suited for a specialized journal in social work, social policy, or psychometric measurement. I would strongly encourage the authors to redirect their submission to a more appropriate venue, where their excellent work will undoubtedly be of great interest to the target readership and have a much greater impact.

Round 2

Reviewer 1 Report

Comments and Suggestions for Authors

Dear authors, I appreciate and value your responses to the initial comments. The changes reflect your willingness to contribute a more solid and better-focused article.
I emphasize the need to include a section that projects your method to a broader context (Globla). How can this study be applied in other countries? In other cities? In other contexts? The language is now more fluid and formal.

Reviewer 4 Report

Comments and Suggestions for Authors

The authors have done a great job of revising their manuscript and have successfully addressed nearly all of the substantive concerns I raised in the initial review. The only remaining concern is the partial disconnect between the justification for the sample composition provided. This, however, is a minor issue that can be easily resolved. 

In the Methods section, please add the important contextual information from your response letter. Specifically, state that under Japan's national guidelines from the Ministry of Health, Labour, and Welfare, the required services and processes related to welfare support are identical for practitioners regardless of their social work certification status. This will fully justify the inclusion of both groups in your sample and resolve the misnomer critique.

Once this minor addition is made, I believe the manuscript will be a valuable contribution to the literature on social work practice and measurement.
